# Pluripotent Cells Expressing APOE4 Exhibit a Pronounced Pro-Apoptotic Phenotype Accompanied by Markers of Hyperinflammation and a Blunted NF-κB Response

**DOI:** 10.3390/ijms26199283

**Published:** 2025-09-23

**Authors:** Wiebke Schulten, Nele Johanne Czaniera, Anna Lena Buschheuer, Antonia Liermann, Axel Wiegand, Barbara Kaltschmidt, Christian Kaltschmidt

**Affiliations:** 1Department of Cell Biology, University of Bielefeld, 33615 Bielefeld, Germany; wiebke.schulten@uni-bielefeld.de (W.S.);; 2Forschungsverbund BioMedizin Bielefeld, Ostwestfalen-Lippe (OWL) (FBMB E.V.), 33615 Bielefeld, Germany; 3Molecular Neurobiology, University of Bielefeld, 33615 Bielefeld, Germany

**Keywords:** *APOE*, Alzheimer’s disease, TNF-α, apoptosis, necroptosis, iPSC, NF-κB, double strand breaks, DNA damage, hyperinflammation

## Abstract

Alzheimer’s disease (AD) is a progressive neurodegenerative disorder that poses an increasing burden on society. It is characterized by the presence of neurofibrillary tangles (NFTs) and amyloid-beta (Aβ) plaques. AD is a multifactorial disease, with one of the strongest genetic risk factors being the *APOE4* allele. In this study, we investigated the impact of APOE4 on NF-κB signaling in induced pluripotent stem (iPS) cells. Our results indicate that APOE4 may influence the subcellular localization of the pluripotency marker OCT4, showing a predominantly nuclear localization in *APOE4* cells, whereas it appears cytoplasmic in *APOE3* cells. Additionally, NF-κB activation via its canonical subunits is blunted in *APOE4* cells. Interestingly, *APOE4* cells still exhibit increased transcription of key hyperinflammatory markers *CCL2*, *CXCL10* and *COX2*, which are known NF-κB target genes, and exhibit a significantly higher rate of apoptosis compared to *APOE3* cells—independent of TNF-α stimulation. Moreover, an elevated incidence of DNA double-strand breaks was observed in *APOE4* cells. However, the precise molecular mechanisms by which APOE4 suppresses NF-κB activation while simultaneously promoting inflammation and apoptosis remain unclear. Further research is required to elucidate these underlying pathways.

## 1. Alzheimer’s Disease

Alzheimer’s disease was first described by the German psychiatrist and neuropathologist Alois Alzheimer, who, during his presentation at the 37th Conference of Southwest German Psychiatrists in Tübingen in 1906, reported the presence of plaques and NFTs in the brain of a long-term psychiatric patient, based on histopathological examination [1]. The hallmark amyloid plaques primarily consist of Aβ peptides—proteolytic fragments of 40 or 42 amino acids—produced through sequential cleavage of the amyloid precursor protein (APP) by β- and γ-secretases. Tau, a microtubule-associated protein, plays a key structural role in maintaining axonal integrity by stabilizing microtubules. In AD, tau becomes abnormally redistributed to the somatodendritic compartment, where it undergoes hyperphosphorylation—potentially mediated by kinases such as protein kinase A (PKA) [2]. Hyperphosphorylated tau tends to aggregate, forming NFTs that disrupt axonal transport and may exacerbate extracellular Aβ accumulation [2]. According to the Thal amyloid staging framework, amyloid plaques—representing extracellular deposits of Aβ—are primarily observed in advanced stages of AD pathology [3]. A prominent feature of AD progression is chronic neuroinflammation, characterized by prolonged activation of microglia, the brain’s resident macrophages, and other immune cells. This sustained immune activation contributes to the worsening of both amyloid and tau pathologies. Mounting evidence suggests that neuroinflammation serves as a crucial driver in the progression of AD [4]. Within this inflammatory context, the proinflammatory cytokine tumor necrosis factor-alpha (TNF-α) has emerged as a key mediator of neuroinflammation in AD. This is supported by clinical observations that anti-TNF therapies—such as TNF receptor fusion proteins like Enbrel—may reduce AD risk [5].

## 2. NF-κB

The transcription factor NF-κB serves as a central regulator of neuroinflammation and is critically involved in a broad spectrum of cellular functions within the mammalian nervous system [6]. In the nervous system, NF-κB can be activated by various extracellular stimuli, including amyloid-beta [7], the proinflammatory cytokine TNF-α [8,9], the excitatory neurotransmitter glutamate [10,11], and glutamatergic agonists such as kainate [11] and N-methyl-D-aspartate (NMDA) [12]. The human NF-κB family comprises five DNA-binding subunits: REL (cRel), RELA (p65), RELB (RelB), NFKB1 (p50), and NFKB2 (p52), of which NFKB1 and NFKB2 lack intrinsic transactivation domains [13]. NF-κB signaling is typically divided into three branches: the canonical, non-canonical, and atypical pathways. Both the canonical and non-canonical routes rely on the inhibitor of κB (IκBα) and the IκB kinase (IKK) complex. In non-canonical signaling, ligand binding to receptors such as CD40 leads to activation of IKK1 through NF-κB-inducing kinase (NIK), resulting in phosphorylation-dependent processing of the p100 precursor into p52 and subsequent nuclear translocation of the p52/RELB heterodimer [14]. The transcriptional activity of NF-κB follows temporally distinct programs, including constitutive, inducible, and subunit-specific patterns, each of which may be activated or suppressed across extended timescales [15]. The canonical NF-κB signaling cascade is initiated by ligand-induced trimerization of TNFR1 through TNF-α binding [16], which facilitates the assembly of an intracellular complex involving adaptor proteins such as TNFR1-associated death domain (TRADD), receptor-interacting protein kinase 1 (RIPK1), and E3 ubiquitin ligases including TNF receptor-associated factor 2 (TRAF2) and cellular inhibitors of apoptosis proteins 1 and 2 (cIAP1/2) [17].

## 3. Apolipoprotein E

AD is predominantly a genetic disorder, with heritability estimates ranging from 60% to 80% [18]. Among the most significant genetic risk factors is the presence of the *apolipoprotein E4* (*APOE4*) allele [19]. Compared to other *APOE* alleles such as *E2* and *E3*, carriers of the *E4* variant exhibit an approximately 40% increased risk of developing certain neurodegenerative diseases [20]. Three major *APOE* isoforms—*APOE2*, *APOE3*, and *APOE4*—are defined by two single nucleotide polymorphisms (SNPs). APOE3, the most common isoform, contains a cysteine at position 112 and an arginine at position 158. APOE2, the least prevalent, has cysteines at both positions, while APOE4 features arginines at both residues. These amino acid substitutions influence the protein’s tertiary structure: in APOE4, a salt bridge forms between the N-terminal and C-terminal domains, resulting in a compact conformation not seen in the more flexible structures of APOE2 or APOE3 [21]. Functionally, APOE4 exhibits reduced lipidation efficiency compared to APOE3, impairing lipid transport between glial cells and neurons. This deficiency compromises neuronal homeostasis and viability and may be a major driver of AD pathogenesis [22].

In this study, we compared the *APOE4* genotype with isogenic *E3*-iPS cells. Our findings revealed a significantly higher rate of apoptosis in *E4* cells, which could not be further increased by TNF-α treatment. This prompted us to examine the TNF-α-induced NF-κB signaling response, where we observed a diminished activation and reduced expression of NF-κB transactivating subunits in *E4* cells relative to *E3* controls. Furthermore, the induction of NF-κB target genes in response to TNF-α was also attenuated in *APOE4* cells.

## 4. Results

### 4.1. The APOE4 Allele Variant Shows Molecular Signs of Higher Pluripotency

For this study, induced pluripotent stem cells homozygous for either the *APOE3* allele or the *APOE4* allele were used. First, they were analyzed for specific characteristics of iPS cells. They both showed the typical morphological features of iPSC colonies with round, sharp-edged colonies. The cell culture colonies were sharply defined (Figure 1A) [23]. The pluripotency markers OCT4 and SSEA4 could also be detected at protein level by immunocytochemical staining (Figure 1B), with the *APOE4* cell line showing a visibly stronger OCT4 nuclear localization than in the *APOE3* cells. SSEA4 is cytoplasmatically localized in both populations. At the mRNA level, a similar expression profile of *OCT4*, *MYC* and *SOX2* was observed in both populations (Figure 1C). To verify whether the *APOE3* and *APOE4* genotypes were involved, the corresponding locus was amplified by PCR spanning the SNPs rs429358 and rs7412 and then sequenced (Figure 1D). We could confirm the haplotypes T/C (*APOE3*) and C/C (*APOE4*) in both cell lines. Based on the previous results, we conclude that *APOE4* cells depict more pluripotency markers than *APOE3* cells.

### 4.2. NF-κB Activation Is Blunted in Pluripotent Cells with APOE4 Alleles

Treatment with the cytokine TNF-α over various periods of time showed a visible translocation of the NF-κB subunit RelA into the cell nucleus in *APOE3* cells after 60 min, whereas activation did not occur in *APOE4* cells (Figure 2A). Quantification of the fluorescence intensity in the cell nucleus confirms the observations (Figure 2B). In *APOE3* cells, the RelA signal in the nucleus increases significantly to up to 1.7 times after 90 min of exposure to TNF-α. *APOE4* cells, on the other hand, show no or reduced activation of RelA. Here, the level of the untreated control was never reached after TNF-α induction.

The same can be observed for the cRel subunit (Figure 3A). After 30 min of treatment, the translocation of cRel into the cell nucleus of *APOE3* cells becomes visible and persists up to 90 min of treatment. In *APOE4* cells, however, there was no visible translocation of cRel into the cell nucleus. Quantification supports these observations (Figure 3B). While the fold change in nuclear fluorescence intensity in *APOE3* cells continuously increases to 1.7 times after 90 min, the signal in *APOE4* cells decreases significantly to 0.6 times after 10 min and peaks at 1.1 times after 30 min before decreasing again. These results suggest that NF-κB signal transduction appears to be blunted in *APOE4* cells.

### 4.3. The APOE4 Allele Has a Gene-Specific Influence on the Expression and Induction of NF-κB Target Genes

Next, the gene expression of the NF-κB subunits *RELA*, *RELB*, and *REL* and the well-established target genes *TNFR2*, *IL-1β*, *CCL2*, *CXCL10*, *ICAM1*, and *COX2* was examined with and without stimulation by TNF-α (Figure 4). The expression of NF-κB subunits was significantly lower in *APOE4* cells than in *APOE3* cells (Figure 4A). Nevertheless, analysis of the target genes revealed that the expression of the genes *CCL2*, *CXCL10*, and *COX2* was significantly higher in the untreated state than in *APOE3* cells as quantified by ΔCt calculations (Figure 4B).

Furthermore, we used ΔΔCt calculations to compare inductions of TNF-α in *APOE3* and *APOE4* cells (Figure 4C). The gene for the NF-κB activating *TNFR2* was immediately induced after one hour of TNF-α treatment in *APOE3* cells (more than seven-fold) whereas induction in *APOE4* was only two-fold with no induction after 4 h. In contrast, TNF-α induction of *ICAM1* was more prominent in *APOE4*. As known from the literature, *IL-1B* as a late target gene in *APOE3* cells is no induction in *APOE4*. *CXCL10* gene responded with rapid induction after 1 h and 4 h in *APOE3* cells but was severely blunted in *APOE4* cells with much lower values after 4 h of treatment. Similar picture was evident with the induction of *CCL2*, which was more than ten-fold in *APOE3* cells, but completely blunted in *APOE4* cells. Finally, the *COX2* expression was reduced to half of untreated in *APOE4* cells in contrast to *APOE3* cells.

### 4.4. Cells Harboring the Homozygous APOE4 Genotype Undergo Apoptosis More Frequently than Those with APOE3

Next, we analyzed the role of TNF-α in regulating different types of cell death. We examined apoptosis and necroptosis in both cell lines before and after TNF-α treatment, since these are the most frequent forms of cell death AD [5]. Cleaved caspase-3 was stained as a marker for apoptosis and phosphorylated MLKL as a marker for necroptosis. We found that, regardless of TNF-α treatment, the *APOE4* cells had a significantly higher apoptosis rate than the *APOE3* cells (Figure 5A,B). While the rate in *APOE3* cells remains just under 10% (more precisely 9.46% when untreated), it is twice as high (120 min) to almost six times as high (60 min) in *APOE4* cells. When apoptotic pathways are blocked, necroptosis can serve as a compensatory cell death mechanism [5]. Since the *APOE3* cells did not respond to TNF-α with apoptosis, we looked at necroptosis in the next step. No significant increase in the necroptosis marker pMLKL was observed in either cell line (Appendix A). However, there was a difference in the localization of pMLKL: in *APOE3* cells (Appendix A, upper panel), the fluorescence signal is predominantly cytoplasmic, while in *APOE4* cells (Appendix A, lower panel), it occurs more frequently in the cell nucleus. But, pMLKL is reported to be localized to the cell membrane in cells undergoing necroptosis [24].

### 4.5. Double Strand Break-Bearing Cells Are Mainly Present in APOE4 Genotype

DNA damage and defective DNA repair are hallmarks of aging and several neurodegenerative disorders including AD [25,26]. Thus, we wanted to investigate whether DNA damage plays a role in *APOE* iPS cell lines. For this purpose, we stained the phosphorylated histone H2AX (γH2AX) in the cells. It is considered a marker for DNA double-strand breaks, as it is bound to DNA to reduce genomic instability [27]. The staining shows that more γH2AX is accumulated in cells with the *APOE4* allele than in *APOE3* cells (Figure 6A). However, quantification reveals only a trend that is not statistically significant (Figure 6B). A non-parametric Mann–Whitney U test confirmed this with U = 0, corresponding to a large effect (r = 0.80), although the small sample size (*n* = 3 per group) limits statistical power.

## 5. Discussion

In this study, we investigated the impact of the homozygous *APOE4* genotype in induced pluripotent stem cells compared to isogenic *APOE3* controls. The APOE4 isoform carries two aspartate residues at positions 112 and 158, whereas APOE3 contains a cysteine at position 112 and an aspartate at 158. We sequenced the here used iPS cell models and could confirm the relevant SNPs for both *APOE4* and *APOE3* allels (see Figure 1D). Our data demonstrates that the substitution of cysteine with aspartate at position 112 exerts a profound influence on the expression of inflammatory genes, accompanied by a marked increase in apoptosis. While the effects of APOE4 have been previously examined in iPS cell–derived neurons and astrocytes [28], its role in undifferentiated iPS cells remained unexplored. Here, we address this gap by analyzing APOE4-specific effects in undifferentiated iPS cells. Induced pluripotent stem cells, first described by Yamanaka, serve as an excellent model system for human embryonic stem (ES) cells, while circumventing the associated ethical concerns. iPS cells share key characteristics with ES cells, including the expression of core pluripotency factors and highly similar gene expression profiles [23].

According to Fernandes et al. [29], human induced pluripotent stem cells provide a valuable model for investigating the cellular and molecular mechanisms of late-onset Alzheimer’s disease within the framework of human nervous system development. Rodent models of Alzheimer’s disease have been instrumental in elucidating aspects of AD etiology and pathogenesis. However, mice do not naturally develop amyloid-β plaques or neurofibrillary tangles [30]. In most rodent models, AD-like pathology is artificially induced through mutations in genes associated with familial AD (fAD), such as *App* or *Psen1*. Yet, these mutations account for only approximately 5% of all AD cases (Alzheimer’s Association, 2006). Moreover, mouse APOE shares only ~70% sequence identity with its human ortholog [30]. In humans, the *APOE4* allele represents the most significant genetic risk factor for sporadic AD. Risk varies substantially across populations: in individuals of European descendent, *APOE4* homozygosity increases risk by ~15-fold, while in East Asian populations this effect rises to ~33-fold. In contrast, African American and Hispanic populations exhibit comparatively lower increases in risk, ranging from 1- to 5-fold [31]. Such population-specific variation in genetic risk cannot be recapitulated in rodent models, which lack the genetic diversity inherent to human populations. But, the invention of induced pluripotent stem cell technology has enabled the reprogramming of human somatic cells into pluripotent stem cells and their subsequent differentiation into neural cell types [32]. Human iPSCs (hiPSCs) preserve the donor’s genetic background while erasing most of the acquired epigenetic modifications [33,34,35,36]. This feature provides a unique opportunity to disentangle genetic from epigenetic contributions to age-related brain disorders. To elucidate the developmental differences associated with the *APOE* alleles, a detailed analysis of the undifferentiated iPS cells may be informative.

In this study, we analyzed the morphology and proliferation of *APOE4* and *APOE3* iPS cell lines and observed no notable differences (see Figure 1A). However, we identified differences in the subcellular localization of pluripotency markers such as OCT4. In *APOE4* cells, OCT4 was predominantly localized to the nucleus, whereas in *APOE3* cells, it was more frequently detected in the cytoplasm (see Figure 1B). High levels of OCT4 are known to interfere with differentiation [37].

The proinflammatory cytokine TNF-α is considered one of the key regulators of inflammation in AD, as inhibition of its function using recombinant receptor fusion proteins (e.g., etanercept/Enbrel) and other biologics has been shown to significantly reduce AD incidence [5]. Based on this, we employed TNF-α to activate the NF-κB signaling pathway in iPS cells (see Figure 2 and Figure 3). Unexpectedly, no nuclear translocation of the transactivating NF-κB subunits RelA and cRel was observed in *APOE4* cells, whereas *APOE3* cells exhibited clear nuclear localization. To the best of our knowledge, this is the first report of a single nucleotide polymorphism—resulting in a cysteine-to-aspartate substitution—substantially attenuating NF-κB nuclear translocation. The underlying molecular mechanism remains unresolved and requires further investigation.

In addition, we assessed the expression of NF-κB subunits and downstream target genes in both cell lines before and after TNF-α stimulation (see Figure 4). Remarkably, all three transactivating NF-κB subunits were expressed at less than prox. 50% of the levels observed in *APOE3* cells (see also Figure 4A). A plausible explanation for this observation could be a direct repressive effect of APOE4 on a DNA regulatory element located near the promoters of the affected genes. However, previous studies proposing a DNA-binding function of APOE4 as a transcriptional repressor—specifically in the regulation of *ADAM10*—have not been widely corroborated [38]. Interestingly, UniProt annotations suggest that APOE may act as a negative regulator of gene expression, based on inferred sequence or structural similarities (https://www.uniprot.org/uniprotkb/P02649/entry (accessed on 10 August 2025)). To explore this possibility, we performed a conserved domain search using the NCBI database to identify putative DNA-binding motifs within the APOE protein. However, the citation analysis did not reveal any conclusive evidence for DNA-binding domains.

More recently RNA-seq data revealed transcriptional repressing and transcriptional activation effect specifically for the *APOE4* allele. RNA-seq data from human brain cell types differentiated from iPS cells were pooled and analyzed using co-expression network approaches, resulting in the identification of 857 genes exhibiting expression patterns similar to *APOE*. Promoter analysis of these genes revealed an enrichment of binding motifs for three transcription factors: nuclear factor 1 (NF1), activator protein 1 (AP-1), and nuclear factor κB (NF-κB). Differential expression analysis indicated a greater number of downregulated genes in *APOE4* compared to *APOE3* cells, with 64 in iPSCs, 254 in neurons (including BDNF), 909 in astrocytes, and 1131 in microglia-like cells. In contrast, the largest number of upregulated genes was observed in astrocytes (418). These findings highlight the pronounced cell-type–specific effects of APOE isoforms [28]. But the molecular mechanism by which APOE might exert a negative regulatory effect on NF-κB subunit transcription remains elusive and warrants further investigation.

Furthermore, three out of the six NF-κB target genes analyzed—*CCL2*, *CXCL10*, and *COX2*—were significantly upregulated in *APOE4* iPS cells compared to *APOE3* cells. CCL2 (C-C motif chemokine ligand 2), also known as monocyte chemoattractant protein-1 (MCP-1), is a pivotal chemokine of the CC family that mediates the recruitment of immune cells to the central nervous system, thereby promoting neuroinflammation. Elevated CCL2 expression has been implicated in various neurodegenerative conditions, including AD [39]. CXCL10 (C-X-C motif chemokine ligand 10), a member of the CXC chemokine family, is known to facilitate astrocyte migration and has been shown to colocalize with Aβ deposits in the AD brain [40]. COX2 (cyclooxygenase-2) is a key enzyme in the synthesis of proinflammatory prostaglandins and plays a crucial role in neuroinflammatory processes. In AD, COX2 overexpression has been linked to tau hyperphosphorylation and the formation of neurofibrillary tangles. Given its proinflammatory function, COX2 has been proposed as a therapeutic target; its inhibition may reduce inflammation and slow disease progression [41]. The expression levels of the other three NF-κB target genes—*TNFR2*, *IL-1β*, and *ICAM1*—were not significantly elevated in *APOE4* cells compared to *APOE3* controls. TNFR2 (Tumor Necrosis Factor Receptor 2) is generally considered neuroprotective, in contrast to TNFR1, which exerts proinflammatory effects [42]. In AD, increased expression of TNF-α and TNFR1 has been reported, consistent with their proinflammatory roles, whereas TNFR2 expression is typically downregulated. IL-1β (Interleukin-1 beta) is involved in both acute inflammation and adaptive immune responses [43]. An NF-κB binding site located upstream of its promoter region facilitates enhanced IL-1β expression in response to inflammatory stimuli [44]. The reduced expression of NF-κB subunits observed in *APOE4* iPS cells may impair the transcriptional activation of IL-1β, potentially explaining the lack of increased expression compared to *APOE3* cells. ICAM1 (Intercellular Adhesion Molecule 1) is an inducible cell surface glycoprotein involved in leukocyte adhesion and transmigration during inflammatory responses. It plays a critical role in immune cell trafficking and coordination of immune signaling [45].

In astrocytes differentiated from *APOE4* iPSCs, constitutive NF-κB activity was exhibited, likely due to reduced histone deacetylase activity associated with *TAGLN3* downregulation, leading to a hyperactivated inflammatory state. Consistently, patient-derived *APOE4* astrocytes displayed up to a tenfold stronger transcriptional induction of inflammatory genes, such as *IL-1B* and *CCL2*, in response to proinflammatory stimulation [46].

We next analyzed two major forms of programmed cell death—apoptosis and necroptosis—in both *APOE3* and *APOE4* cell lines. *APOE4* cells exhibited a significantly increased rate of apoptosis compared to *APOE3* controls (Figure 5). In contrast, TNF-α-induced necroptosis levels did not differ significantly between the two genotypes (Appendix A). The reduced expression of NF-κB subunits in *APOE4* cells may lead to downregulation of anti-apoptotic genes, such as *BIRC2/3*, *BCL2L1*, and *TRAF*, which are known to suppress caspase activity [47]. Necroptosis, on the other hand, was not elevated in either cell line following TNF-α treatment (Appendix A). It is well established that necroptosis is predominantly activated when apoptosis is inhibited [48]. Therefore, the absence of increased necroptosis in *APOE4* cells is consistent with this mechanism. Nevertheless, phosphorylated MLKL signals were detected in both cell lines, predominantly localized in the nucleus. Recent studies have demonstrated that nuclear pMLKL may have diverse intracellular roles, including gene regulation, inflammasome activation, and promotion of Ca^2+^ influx [24,48]. Further investigations are required to elucidate the specific mechanisms and functional consequences of necroptosis in these cells.

Finally, we demonstrated that APOE4-expressing cells exhibit a greater number of DNA double-strand breaks (DSBs) compared to APOE3-expressing cells (Figure 6). Previous studies have shown that APOE4 expression is associated with elevated levels of oxidative stress, which contributes to increased DNA damage [49]. One possible mechanistic explanation is that APOE4 contains fewer cysteine residues than APOE3 or APOE2. These cysteine residues serve as binding sites for the neurotoxic lipid peroxidation product 4-hydroxynonenal [50]. Consequently, the reduced availability of cysteine residues in APOE4 may impair the cellular defense against lipid peroxidation.

In summary, our findings indicate that the *APOE4* allele affects both hyperinflammation and apoptosis in pluripotent stem cells. Although NF-κB activation via the subunits RelA and cRel is attenuated, the cells still undergo apoptosis, and the expression of NF-κB target genes remains consistently elevated compared to *APOE3* cells (Figure 7). Additionally, the observed localization of OCT4 suggests a potential impact on the differentiation capacity of these stem cells. The molecular mechanisms of APOE4 mediated repression of NF-κB and the induction of apoptosis remains elusive. Further research is necessary to confirm this hypothesis and establish definitive conclusions.

An additional intriguing aspect concerns the evolutionary role of *APOE4*. It is believed to be the most ancestral isoform of the *APOE* gene, with the *APOE2* and *APOE3* variants emerging later in human evolution and contributing to increased lifespan. Individuals homozygous for the *APOE4* allele are known to have higher total cholesterol levels compared to those carrying the *APOE3* or *APOE2* alleles [51]. Since life expectancy in early human history was relatively short, the detrimental effects of APOE4—such as its association with AD—likely had little impact. From the evolutionary perspective, carriage of the *APOE4* allele may confer certain advantages compared to non-carriers. For instance, women heterozygous for the *APOE4* allele were reported to have 0.3 to 0.5 more offspring than women homozygous for *APOE3*, while *APOE4* homozygous women had 1.4 to 2.1 more children. Furthermore, children carrying the *APOE4* allele tend to exhibit faster growth compared to non-carriers, suggesting a potential developmental advantage [52]. Moreover, elevated lipid levels associated with APOE4 expression have been linked to increased resistance to various pathogens, including Giardia and hepatitis C virus. In high-infection environments, *APOE4* carriers have demonstrated improved cognitive performance relative to non-carriers [53].

Despite the insights gained, this study has certain limitations. It is based on a single iPSC line, as generating an isogenic control line is technically challenging. To address this, we chose to use a well-characterized iPSC line along with its corresponding isogenic control, both obtained from the Jackson Laboratory.

## 6. Materials and Methods

### 6.1. Cell Culture

The homozygous iPS cell lines JIPSC001150 and the corresponding revertant JIPSC001162, from the Jackson Laboratory (Bar Harbor, ME, USA), were used for all experiments performed in this study. Cells were cultured in 6-well plates (Sarstedt, Nümbrecht, Germany) on 5 µg/mL vitronectin (Thermo Fisher Scientific, Waltham, MA, USA) and in E8 Flex medium (Thermo Fisher Scientific, Waltham, MA, USA) according to manufacturer’s guidelines. They were incubated in a humidified incubator (Binder, Tuttlingen, Germany) at 37 °C and 5% CO_2_. Passaging was performed according to the guidelines of the medium manufacturer. Morphology and pluripotency markers were analyzed (see Figure 1).

### 6.2. Genotyping of iPS Cell Lines

The genotyping of the cell lines used was analyzed by Sanger sequencing. For this purpose, a genomic PCR of the *APOE* locus was performed in which the two relevant SNPs rs429358 and rs7412 are located. The primers fwd: 5′-AGCCCTTCTCCCCGCCTCCCACTGT-3′ and rev: 5′-CTCCGCCACCTGCTCCTTCACCTCG-3′ were used for amplification. PCR amplification was initiated with an initial denaturation at 95 °C for 3 min. This was followed by 40 cycles of denaturation at 95 °C for 20 s, annealing at 60 °C for 30 s, and extension at 68 °C for 2 min and 20 s. The protocol ended with a final extension at 68 °C for 5 min. The PCR product was purified as described by the manufacturer using Exo-CIP Rapid PCR Cleanup Kit (New England Biolabs, Frankfurt am Main, Germany).

### 6.3. TNF-α Treatments

For the TNF-α treatments, the cells were seeded on µ-slide 8 well (Ibidi, Gräfelfing, Germany) and cultivated for two days under the conditions described above. Prior to the treatment, the medium was removed, washed once with 1× phosphate-buffered saline (PBS; PAN-Biotech, Aidenbach, Germany) and fresh medium was added to the cells, treated with 10 ng/µL TNF-α (Peprotech, Cranbury, NJ, USA). The treatment was carried out for 10 min, 30 min, 60 min, 90 min, 2 h, 3 h, 4 h, 16 h, 20 h or 24 h. Afterwards, the cells were washed again with 1× PBS and processed for immunocytochemistry as described below.

### 6.4. Immunocytochemistry

To fix the cells for immunocytochemistry (ICC), they were incubated for 10 min with 4% paraformaldehyde at RT. The cells were then washed three times with 1× PBS and the blocking solution, consisting of 1× PBS, 0.02% Triton-X (Sigma-Aldrich, St. Louis, MI, USA) and 5% goat serum (Dianova, Hamburg, Germany), was applied to the cells for 45 min. This was followed by application of the primary antibodies (see Table 1) for 1 h at RT. Before the secondary antibody was applied to the cells, the cells were washed three times with 1× PBS. Alexa fluor 555 (goat anti-rabbit) and Alexa Fluor 488 (goat anti-mouse; both Thermo Fisher Scientific, Waltham, MA, USA) were used as secondary antibodies at a dilution of 1:300. Mounting medium with DAPI was used as nuclear stain (Ibidi, Gräfelfing, Germany). Imaging was performed on a confocal laser scanning microscope (Leica STELLARIS 8 FALCON; Leica Microsystems GmbH, Wetzlar, Germany), further processing was performed with ImageJ 1.54p (National Institutes of Health) and CorelDRAW 2023 (Corel Corporation, Ottawa, ON, Kanada) and statistical analysis with GraphPad Prism 8.3.0 (Graphpad Software, Inc., Boston, MA, USA).

### 6.5. qPCR

The RNA of the cells was isolated using the NucleoSpin^®^ RNA kit (Macherey-Nagel, Düren, Germany) and then transcribed into cDNA using the First Strand cDNA Synthesis Kit (Thermo Fisher Scientific, Waltham, MA, USA). For all qPCR reactions the qPCRBIO SyGreen Mix (PCR Biosystems, London, UK; Cat. No. PB20.14-05) was used. For each reaction, triplicates were prepared in the Rotor Gene 6000 (QIAGEN, Hilden, Germany). The primer sequences can be found in Table 2. GAPDH and RPLP0 served as housekeeping genes. LinRegPCR [54,55], Microsoft Excel (Microsoft, Redmond, Washington, DC, USA) and finally GraphPad Prism 8.3.0 were used for the analysis.

### 6.6. Nomenclature

Human gene names were used as gene symbols provided by the HUGO database (https://www.genenames.org/ (accessed on 5 August 2025)). Protein names were used as provided by Uniprot (https://www.uniprot.org/ (accessed on 5 August 2025)).

## Figures and Tables

**Figure 1 ijms-26-09283-f001:**
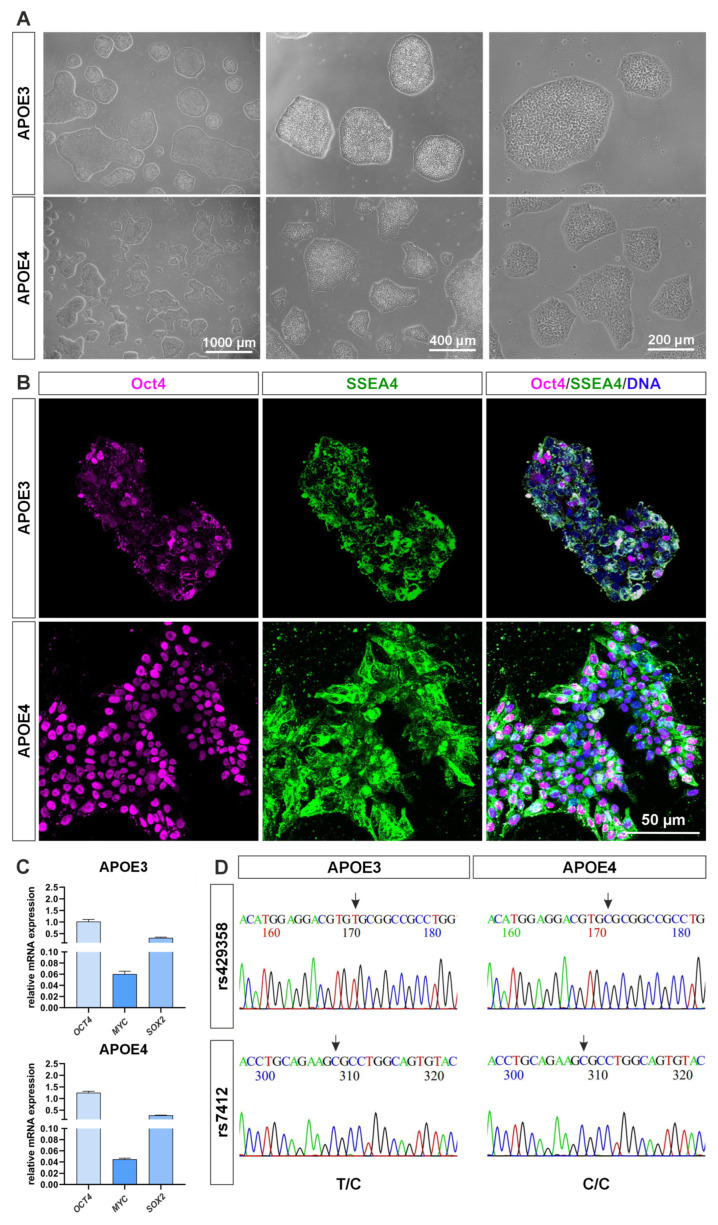
Characterization of *APOE* iPS cells. (**A**) Phase contrast images of successfully cultured induced pluripotent stem cell lines homozygous for either the *APOE3* or *APOE4* allele. (**B**) Immunocytochemical staining of the pluripotency markers OCT4 and SSEA4 confirms expression of both markers in each cell line, validating their pluripotent state. (**C**) Relative mRNA expression levels of key pluripotency-associated genes *OCT4*, *MYC*, and *SOX2*. (**D**) Genotyping analysis of the *APOE* locus, covering the relevant single nucleotide polymorphisms rs429358 and rs7412.

**Figure 2 ijms-26-09283-f002:**
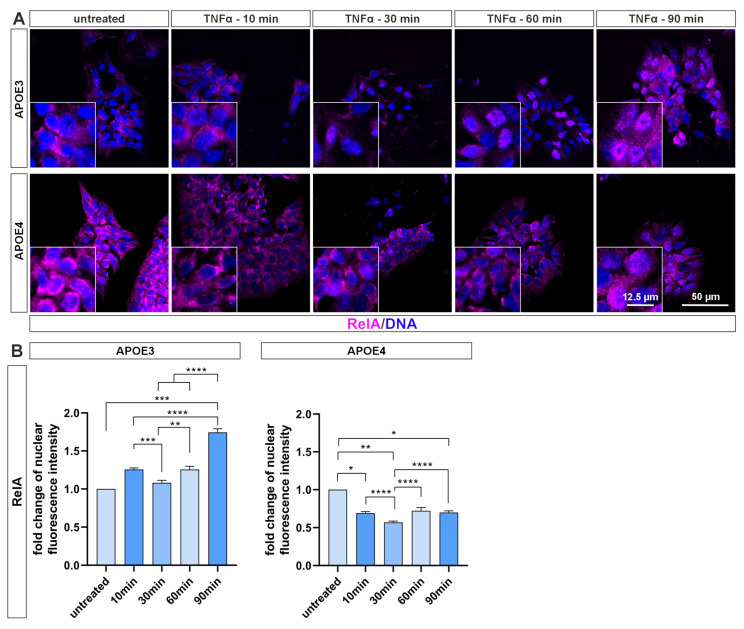
Blunted NF-κB RelA activation in *APOE4* iPS cells following TNF-α stimulation. (**A**) Immunocytochemical staining of NF-κB subunit RelA in *APOE* iPS cells after stimulation with TNF-α for various time points. A clear nuclear translocation of the subunit is observed in *APOE3* cells, whereas *APOE4* cells exhibit no detectable activation. (**B**) Quantification of nuclear localization intensities using ImageJ (NIH, Bethesda, MD, USA) confirms significant activation of RelA in *APOE3* cells, whereas a marked reduction in *APOE4* cells was measured. Statistical analysis was performed using one-way ANOVA (*n* = 3), followed by multiple comparisons corrected using the False Discovery Rate (FDR) method of Benjamini, Krieger, and Yekutieli. * q ≤ 0.05, ** q ≤ 0.01, *** q ≤ 0.001, **** q ≤ 0.0001. Data are presented as mean ± standard error of the mean.

**Figure 3 ijms-26-09283-f003:**
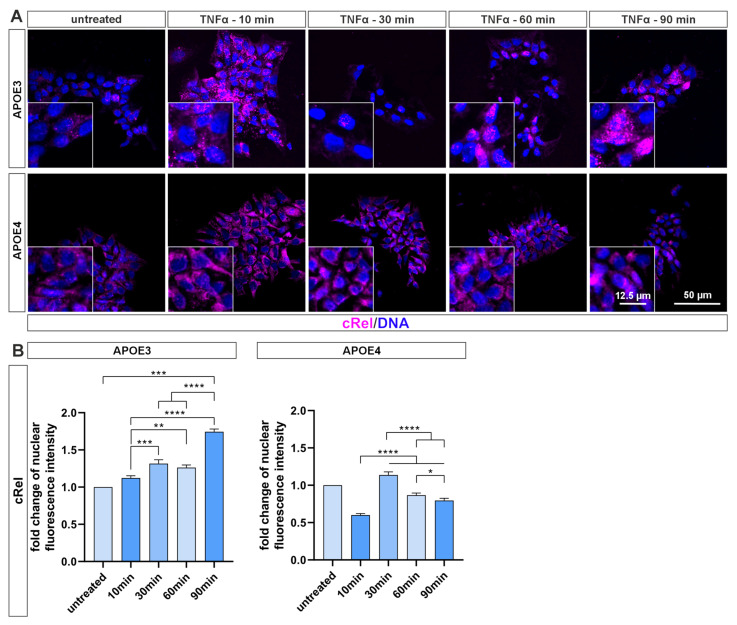
Blunted NF-κB cRel activation in *APOE4* iPS cells following TNF-α stimulation. (**A**) Immunocytochemical staining of NF-κB subunit cRel in *APOE* iPS cells after stimulation with TNF-α for various time points. A clear nuclear translocation of the subunit is observed in *APOE3* cells, whereas *APOE4* cells exhibit no detectable activation. (**B**) Quantification of nuclear localization intensities using ImageJ (NIH, Bethesda, MD, USA) confirms significant activation of cRel in *APOE3* cells, whereas a marked reduction in *APOE4* cells was measured. Statistical analysis was performed using one-way ANOVA (*n* = 3), followed by multiple comparisons corrected using the False Discovery Rate (FDR) method of Benjamini, Krieger, and Yekutieli. * q ≤ 0.05, ** q ≤ 0.01, *** q ≤ 0.001, **** q ≤ 0.0001. Data are presented as mean ± standard error of the mean.

**Figure 4 ijms-26-09283-f004:**
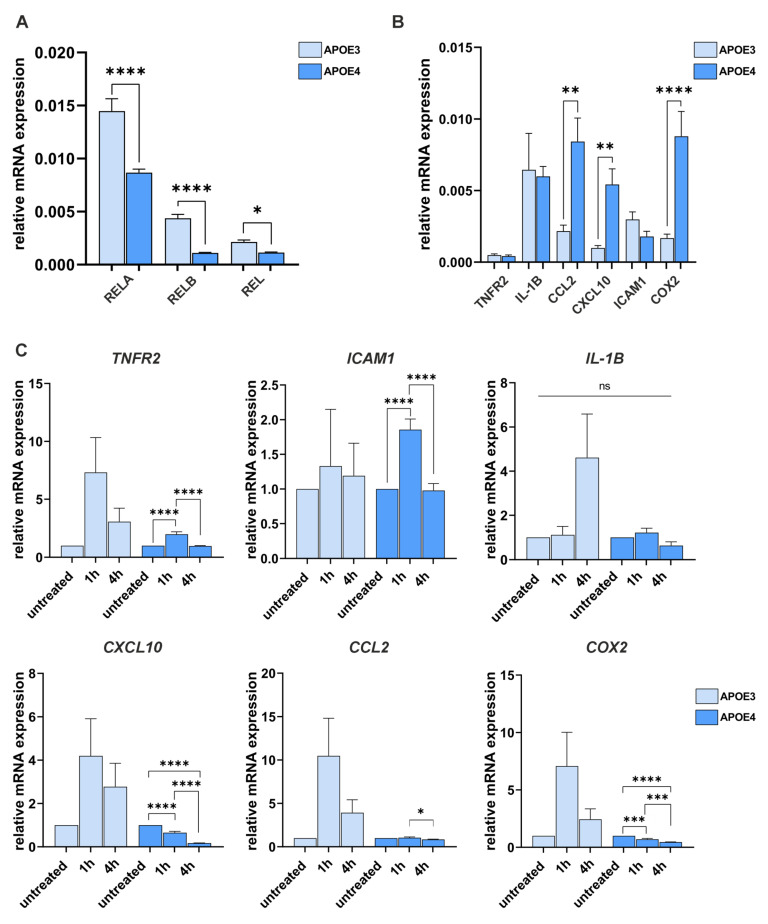
Differential expression of NF-κB subunits and target genes in *APOE* iPS cells before and after TNF-α stimulation. (**A**) RT-qPCR analysis reveals significantly reduced basal expression levels of the NF-κB subunits *RELA*, *RELB*, and *REL* in *APOE4* cells compared to *APOE3* cells. (**B**) In the absence of TNF-α stimulation, *APOE4* cells exhibit significantly elevated expression of selected NF-κB target genes, including *CCL2*, *CXCL10*, and *COX2*. (**C**) Following TNF-α treatment, a robust transcriptional activation of NF-κB target genes is observed in *APOE3* cells. In contrast, *APOE4* cells show a gene-specific activation pattern (*TNFR2*, *ICAM1*), or even a profound downregulation of gene expression (*CXCL10*, *CCL2*, *COX2*). Statistical analysis was performed using one-way ANOVA (*n* = 6), followed by multiple comparisons corrected using the False Discovery Rate (FDR) method of Benjamini, Krieger, and Yekutieli. ns = not significant; * q ≤ 0.05, ** q ≤ 0.01, *** q ≤ 0.001, **** q ≤ 0.0001. Data are presented as mean ± standard error of the mean.

**Figure 5 ijms-26-09283-f005:**
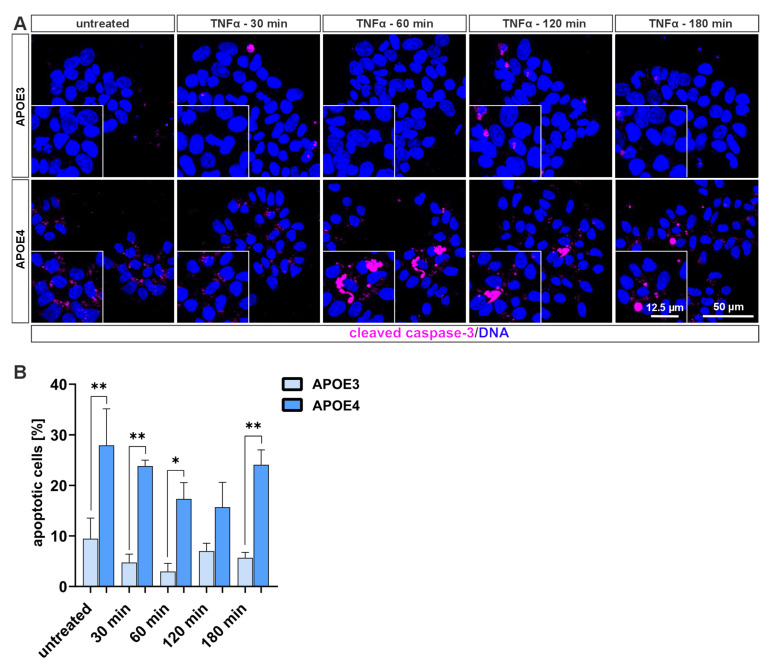
Analysis of the cell death mechanism apoptosis in *APOE* cells. (**A**) Immunocytochemical staining for cleaved caspase-3, a marker of apoptosis, was performed following TNF-α treatment. *APOE4* cells exhibited visibly higher levels of cleaved caspase-3 compared to *APOE3* cells. (**B**) Quantification of the apoptotic response showed a significantly higher apoptosis rate in *APOE4* cells relative to *APOE3*. Statistical analysis was performed using one-way ANOVA (*n* = 3), followed by multiple comparisons corrected using the False Discovery Rate (FDR) method of Benjamini, Krieger, and Yekutieli. * q ≤ 0.05, ** q ≤ 0.01. Data are presented as mean ± standard error of the mean.

**Figure 6 ijms-26-09283-f006:**
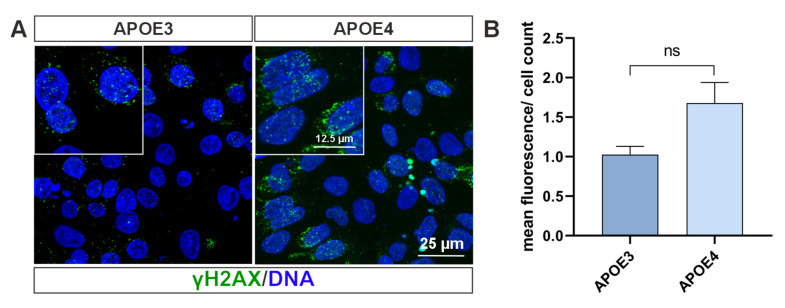
DNA damage in *APOE* cells. (**A**) Immunocytochemical detection of the DNA double-strand break marker γH2AX in *APOE* cells reveals elevated γH2AX levels in *APOE4* cells compared to those expressing *APOE3*. (**B**) Quantification of mean cellular fluorescence supports this observation. Statistical analysis was performed using the Mann–Whitney U test (*n* = 3). ns = not significant. Data are presented as mean ± standard error of the mean.

**Figure 7 ijms-26-09283-f007:**
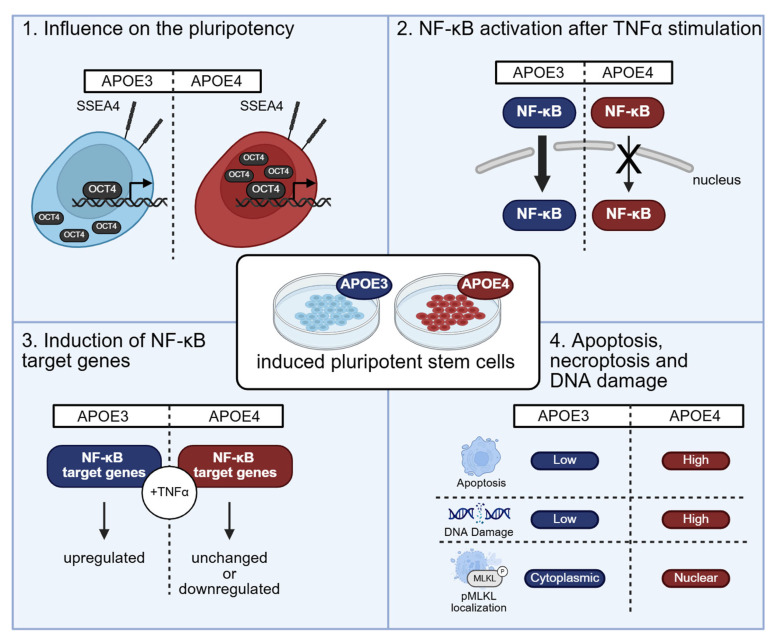
Summary of the impact of the *APOE* allels on NF-κB signaling in induced pluripotent stem cells. The *APOE4* allele affects the pluripotency of iPSCs, as indicated by predominant nuclear localization of the transcription factor OCT4, in contrast to its distribution in *APOE3* cells. NF-κB activation via the subunits RelA and cRel is attenuated in *APOE4* cells after TNF-α treatment, resulting in unchanged or downregulated expression of NF-κB target genes, whereas these genes are then upregulated in *APOE3* cells. Furthermore, *APOE4* cells exhibit increased levels of apoptosis and DNA double-strand breaks compared to *APOE3* cells. Notably, pMLKL localization also differs between genotypes, with cytoplasmic distribution observed in *APOE3* cells and a more nuclear localization in *APOE4* cells.

**Table 1 ijms-26-09283-t001:** Primary antibodies used for ICC in this study.

Primary Antibody	Host	Dilution	Manufacturer
SSEA4	Mouse	1:100	Thermo Fisher Scientific, Waltham, MA, USA
OCT4	Rabbit	1:200	Thermo Fisher Scientific, Waltham, MA, USA
RelA	Rabbit	1:400	Cell Signaling Technology, Danvers & Boston, MA, USA
RelB	Rabbit	1:100	Cell Signaling Technology, Danvers & Boston, MA, USA
cRel	Rabbit	1:100	Cell Signaling Technology, Danvers & Boston, MA, USA
pMLKL	Rabbit	1:200	Cell Signaling Technology, Danvers & Boston, MA, USA
cleaved caspase-3	Rabbit	1:400	Cell Signaling Technology, Danvers & Boston, MA, USA
γH2AX	Mouse	1:500	Thermo Fisher Scientific, Waltham, MA, USA

**Table 2 ijms-26-09283-t002:** Primer used for RT-qPCR.

Primer	Sequence (5′ to 3′)
*GAPDH* fwd	CATGAGAAGTATGACAACAGCCT
*GAPDH* rev	AGTCCTTCCACGATACCAAAGT
*RPLP0* fwd	TGGTCATCCAGCAGGTGTTCGA
*RPLP0* rev	ACAGACACTGGCAACATTGCGG
*TNFR2* fwd	CGTTCTCCAACACGACTTCATCC
*TNFR2* rev	ACGTGCAGACTGCATCCATGCT
*IL-1B* fwd	ATGATGGCTTATTACAGTGGCAA
*IL-1B* rev	GTCGGAGATTCGTAGCTGGA
*CCL2* fwd	CAGCCAGATGCAATCAATGCC
*CCL2* rev	TGGAATCCTGAACCCACTTCT
*CXCL10* fwd	GGTGAGAAGAGATGTCTGAATCC
*CXCL10* rev	GTCCATCCTTGGAAGCACTGCA
*ICAM1* fwd	ATGGCAACGACTCCTTCTCG
*ICAM1* rev	GCCGGAAAGCTGTAGATGGT
*COX2* fwd	CGGTGAAACTCTGGCTAGACAG
*COX2* rev	GCAAACCGTAGATGCTCAGGGA

## Data Availability

Raw data are available upon request from the corresponding author.

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
