# Peer review of "Pluripotent Cells Expressing APOE4 Exhibit a Pronounced Pro-Apoptotic Phenotype Accompanied by Markers of Hyperinflammation and a Blunted NF-κB Response"

_ijms, 2025, doi:10.3390/ijms26199283_

Round 1
Reviewer 1 Report
Comments and Suggestions for Authors
This is a study of the APO3 and APOE4 genotypes in human cells. The cells are induced pluripotential cells (iPSC's) from a cell bank (Jackson), and the iPSC's were derived from fibroblasts of late middle aged men (I think they were also all Caucasian). Thus, the cell lines studied were biased by gender and potentially by race (not sure about this). Overall, the study is interesting and will be of substantial interest to the Alzheimer's disease (AD) research community. APOE4 alleles when homozygous are considered to be risk factors for developing AD, and the authors' paper is relevant b/c of that observation.
The authors carried out experiments about inflammation and apoptosis. The Figures of their data are as follows:
- Figure 1. OK as is. The confocal images are particularly compelling.
- Figure 2. The confocal images of RelA and crew moving into nuclei after TNF-alpha treatment are not compelling. Relative fluorescence across separate images iis not quantitative. In addition, the transfer into nuclear space in the case of RelA/APOE3 at 90 min is not compelling, mainly because the nuclei are not stained well. The authors need to show better images. Also, for the bar charts in (C) and (D), what were the N's? There is no mention of N's in either the figure legend or Methods.
- Figure 3. Bar charts are nicely done, but again there is no mention of N's analyzed in either the legend of Methods.
- Figure 4. Confocal images do not support the bar charts of both cleaved cascade3 (any thoughts about upstream activation of cascades that led to cleavage of the "executioner" cascade3?) and phosphorylated MLKL (for necroptosis). Need much more compelling confocal images (or move them to Supplementary Figure) and indication of N for the bar charts.
- Figure 5. Image is very good, again need N's for bar charts.
- Figure 6. Good educational summary figure.
Overall, this is a very well done study and very well written paper. With the above corrections, it can be published.
Reviewer 2 Report
Comments and Suggestions for Authors
The manuscript addresses a highly relevant question in Alzheimer’s disease research, namely the impact of the APOE4 genotype on NF-κB signaling, apoptosis, and inflammatory pathways, using induced pluripotent stem cells (iPSC). The study is scientifically well-motivated, and the data are presented in a clear and well-structured manner, including both cellular phenotype characterization and functional analyses of NF-κB, apoptosis, and genotoxic stress. The results provide original insights, particularly by highlighting the apparent discrepancy between reduced NF-κB nuclear translocation and the persistence of a hyper-inflammatory profile.
Suggestions for improvement:
- The manuscript is well-written, but alignment with the journal’s style guidelines would strengthen consistency.
- Abbreviations should be defined only at first mention and used consistently in their short form thereafter.
- Some results (e.g., γH2AX) are described as “trends” without statistical significance. Including effect sizes and statistical power would add clarity.
- Certain figures (e.g., Fig. 2 and 4) appear overly dense; simplification or repositioning could improve readability.
- A clearer discussion on how iPSC-based findings may translate to human brain pathophysiology would enhance the manuscript’s impact.
- Gene and protein nomenclature should be standardized according to HGNC/UniProt guidelines (e.g., “OCT4” instead of “oct4”).
- Regarding statistical analysis, only the Mann–Whitney U test is mentioned. If multiple comparisons were performed (especially in qPCR across several genes and time points), clarification on whether corrections (e.g., Bonferroni, FDR) were applied is necessary.
- In the Discussion section, the statement “APOE4 abolishes NF-κB nuclear translocation” is too strong given the single-cell model used. A more cautious formulation such as “substantially attenuates” or “markedly reduces” would be appropriate.
- Integrating a comparison with other similar studies using undifferentiated iPSCs could provide valuable context.
- In Figures 2–5, there is an inconsistency between the text and figure labeling: three significance thresholds (0.05, 0.01, 0.001) are listed, but only the strictest is marked with an asterisk. Standardization is recommended: either simply state “p < 0.05 was considered statistically significant”, or explicitly adopt the conventional notation *“*p ≤ 0.05; **p ≤ 0.01; **p ≤ 0.001”.
Round 2
Reviewer 1 Report
Comments and Suggestions for Authors
This is a paper on human induced pluripotential stem cells (iPSC's) that are homozygous for either APOE3 or APOE4 isozymes. The cells were acquired from a commercial source and have been studied in other scientific reports.
The authors have adequately addressed all of my concerns, and most specifically have added the "N's" to their Figures. They have also added several paragraphs of text, mainly in the Discussion section, that enhance their findings.
Thus, I feel that this paper is now ready for publication.